# Molecular Oncology of Bladder Cancer from Inception to Modern Perspective

**DOI:** 10.3390/cancers14112578

**Published:** 2022-05-24

**Authors:** Soum D. Lokeshwar, Maite Lopez, Semih Sarcan, Karina Aguilar, Daley S. Morera, Devin M. Shaheen, Bal L. Lokeshwar, Vinata B. Lokeshwar

**Affiliations:** 1Department of Urology, Yale University School of Medicine, New Haven, CT 06520, USA; soum.lokeshwar@yale.edu; 2Departments of Biochemistry and Molecular Biology, Medical College of Georgia, Augusta University, 1410 Laney Walker Blvd., Augusta, GA 30912, USA; mailopez@augusta.edu (M.L.); semih.sarcan@student.uni-luebeck.de (S.S.); kaaguilar@augusta.edu (K.A.); dmorera@augusta.edu (D.S.M.); 3Department of Urology, University Hospital Schleswig-Holstein, Campus Lübeck, 23562 Lübeck, Germany; 4Yale School of Nursing, Yale University, New Haven, CT 06520, USA; devin.shaheen@yale.edu; 5Georgia Cancer Center, Medical College of Georgia, Augusta University, 1410 Laney Walker Blvd., Augusta, GA 30912, USA; 6Research Service, Charlie Norwood VA Medical Center, Augusta, GA 30904, USA

**Keywords:** molecular oncology, bladder cancer stem cells, lineage plasticity, intra-tumor heterogeneity, urine biomarkers, genomic/transcriptomic profiling, molecular subtypes, prognostic markers

## Abstract

**Simple Summary:**

Bladder cancer (BC) is a common cancer that causes high morbidity and mortality among affected patients. As a carcinogen-driven cancer, molecular oncology of BC has contributed to several fundamental concepts in cancer biology. Molecular insights into the divergent pathways for low- and high-grade BC development, heterogeneity in bladder tumors regarding recurrence and progression, genetic polymorphism, and the associated risk for developing BC among smokers, driver mutations, and molecular subtyping of muscle-invasive BC for prognostic predictions find commonality with other cancers and therefore, have advanced the field of molecular oncology. Similarly, research into lineage plasticity and tumor heterogeneity has revealed both challenges and opportunities in applying molecular approaches to patient care. The quest for finding a “PSA” for BC has contributed to novel technologies in the biomarker field. This review critically assesses and offers a perspective on the areas where molecular oncology could directly contribute to improving patient care.

**Abstract:**

Within the last forty years, seminal contributions have been made in the areas of bladder cancer (BC) biology, driver genes, molecular profiling, biomarkers, and therapeutic targets for improving personalized patient care. This overview includes seminal discoveries and advances in the molecular oncology of BC. Starting with the concept of divergent molecular pathways for the development of low- and high-grade bladder tumors, field cancerization versus clonality of bladder tumors, cancer driver genes/mutations, genetic polymorphisms, and bacillus Calmette-Guérin (BCG) as an early form of immunotherapy are some of the conceptual contributions towards improving patient care. Although beginning with a promise of predicting prognosis and individualizing treatments, “-omic” approaches and molecular subtypes have revealed the importance of BC stem cells, lineage plasticity, and intra-tumor heterogeneity as the next frontiers for realizing individualized patient care. Along with urine as the optimal non-invasive liquid biopsy, BC is at the forefront of the biomarker field. If the goal is to reduce the number of cystoscopies but not to replace them for monitoring recurrence and asymptomatic microscopic hematuria, a BC marker may reach clinical acceptance. As advances in the molecular oncology of BC continue, the next twenty-five years should significantly advance personalized care for BC patients.

## 1. Introduction

Bladder cancer (BC) is a common cancer in the United States with high morbidity and mortality among affected patients [1,2,3]. Since the first true description of BC almost 200 years ago, the diagnosis and management of BC have significantly evolved [4]. In the last forty years, several seminal discoveries related to BC have distinctly advanced the field of molecular oncology [5,6,7,8]. Field cancerization versus clonality of bladder tumors as the basis for BC recurrence, identification of cancer-initiating genes by whole-organ genome mapping, the molecular basis for multifocality and progression, genetic polymorphism, and the risk for developing BC among smokers, and bacillus Calmette-Guérin (BCG) as one of the early forms of immunotherapy are some of the early conceptual contributions that BC research has made to the field of molecular oncology [9,10,11,12,13,14,15,16,17,18,19,20]. Starting with the hematuria screening trials conducted in the 1990s, BC has been at the forefront of developing biomarkers/tests for non-invasively diagnosing cancer, for monitoring its recurrence and progression, and for predicting clinical outcome [9,13,15,16,17,21,22,23,24]. The hundreds of urine biomarkers and some FDA-approved tests have not only resulted in technological advances in molecular diagnostics but have also been early contributors to the concept of “liquid biopsy for molecular testing” [25]. Along with numerous prognostic markers and investigations into the molecular subtypes, BC research has significantly promoted the promise of personalized medicine for cancer patients. BC stem cells, urothelial cell lineage plasticity, and tumor heterogeneity influencing genetic-profiling-based prognostic predictions are some of the other fundamental concepts in BC research that have advanced the field of molecular oncology [26,27,28].

## 2. Overview on BC

There were more than 83,730 new cases of BC in the United States in 2021, with an estimated 17,200 deaths. Approximately 75% of new BC cases occur in males and 25% in females (64,280 and 19,450 respectively) [29]. BC serves as an umbrella term for neoplasms affecting the bladder, with the most common type being urothelial cell carcinoma (UCC) or transitional cell carcinoma (TCC), comprising greater than 90% of BC cases [30]. In other parts of the world where schistosomiasis, a parasitic infection, is endemic, squamous cell carcinoma of the bladder is more common [31]. Additionally, as the inner lining of the renal pelvis and the ureter consists of the urothelium, the malignancies affecting these organs belong to the same category and are called “upper tract urothelial carcinoma” (UTUC) and may share the same biology and molecular pathways.

There are multiple risk factors for BC. The most recognized risk factor for BC is tobacco smoking, with a four-time higher risk of BC in current smokers [32]. Other associated risk factors for BC include industrial chemical exposures, obesity, radiation exposures, and schistosomiasis infections [33]. BC also has a known genetic component with a two-fold increase in risk amongst first-degree relatives [34].

The median age of presentation in patients with BC is 73 years [35]. BC most commonly presents as painless hematuria either through visible gross hematuria or microhematuria on urinalysis. Less commonly, BC can cause lower urinary tract symptoms [36]. Urinary voiding symptoms typically occur due to tumor location near the bladder neck or through mass effect. Diagnosis is generally made through cystoscopic evaluation. The American Urological Association recommends evaluation through formal urinalysis for patients presenting with microhematuria. For patients older than 35 years of age, cystoscopy should be performed in the workup of patients with painless hematuria. Multiphasic CT scans, including delayed phase imaging, are also recommended to aid in diagnosis [37]. 

## 3. Clinical Management of BC

The disparate histological, molecular, and clinical behaviors of low-grade and high-grade tumors, with complexities added by mixed histologies, indicate that BC is not a single disease but rather a collective group of different types of tumors which happen to arise within the bladder. Low-grade bladder tumors remain confined to the urothelium (stage Ta) and rarely invade the lamina propria (stage T1). Carcinoma in situ is a high-grade flat tumor confined to the urothelium (Tis or CIS). Stages Ta, T1, and Tis are referred to as non-muscle-invasive bladder cancer (NMIBC; Figure 1 and Figure 2). High-grade bladder tumors have high invasive potential. Approximately 25% of patients with BC present with a tumor invading the muscle layer of the bladder wall (T2), perivesical fat (stage T3), or the adjacent organs (stage T4; Figure 1) [38]. Stages ≥ T2 are defined as muscle-invasive bladder cancer (MIBC; Figure 2). Patients with MIBC have a ≥50% chance of developing metastasis and have a 50% chance of survival at five years.

Patients with low-grade bladder tumors are managed with transurethral resection of the tumor and surveillance. Patients with high-grade NMIBC most often receive mostly intravesical BCG following tumor resection. Despite complete tumor resection, new tumors may arise in the bladder, clinically referred to as a recurrence. Given that BC is carcinogen-driven cancer, both field cancerization and clonal expansion have been documented in the case of BC recurrence [39,40]. As was reported nearly forty years ago, bladder tumors demonstrate variable recurrence and progression rates [41,42]. Tumor recurrence rates of about 50% have been reported for low-grade tumors, most of which are stage Ta and have a low progression rate [41,43,44]. Recurrence rates of 57%, 67%, and 77%, and 5-year progression rates of 7%, 26%, and 46% have been reported for low, intermediate, and high-risk patients with NMIBC, respectively [45]. In this study, 92% of patients had high-grade NMIBC and underwent BCG treatment. In high-risk high-grade BC, 74.3% recurrence rate has been reported [46].

The main treatment for MIBC is cystectomy with pelvic lymphadenectomy. Neoadjuvant chemotherapy is intended for down-staging of the disease and for treating micrometastases prior to cystectomy [47]. Patients with the metastatic disease receive systemic treatment, although consolidative extirpative surgery or cystectomy in node-positive patients may offer some benefit [48,49]. Immunotherapy regimens have been approved as both first and second-line treatments for metastatic disease, however, the response rate is ≤25% and chemotherapy in the adjuvant and salvage setting remains the standard of care [50,51,52]. Chemotherapy regimens for BC are largely cisplatin-based. Further, gemcitabine plus cisplatin is a preferred chemotherapy regimen compared to MVAC (methotrexate, vinblastine sulfate, doxorubicin hydrochloride (Adriamycin), and cisplatin) due to its better tolerability in the neoadjuvant and adjuvant/salvage settings [51,53,54,55,56,57,58]. Despite chemotherapy, the median survival of patients with metastatic disease is less than two years due to treatment failure. This narrative shows that the clinical management of BC requires complex decision-making amongst physicians and patients to optimize a personalized treatment plan.

## 4. Molecular Basis of BC and Clinical Translation

The disparate behaviors of low-grade tumors, which almost always are NMIBC, and high-grade tumors, which are invasive and become MIBC, have driven the experimental advances that led to several key concepts, not just in BC but also in cancer research in general. These include the cellular origin of low- and high-grade tumors, urothelial/BC stem cells, tumor heterogeneity, lineage plasticity, influence of tumor microenvironment, genetic landscaping and driver mutations, “omic-based” profiling (mainly transcriptomic), and the molecular subtypes. The recurrent nature of bladder tumors, the availability of matched transurethral resection and cystectomy specimens, exfoliated urothelial cells, and the ease of assaying free-circulating DNA, transcripts, and soluble proteins/metabolites in urine have all enabled such investigations. These studies have certainly advanced knowledge about the biology of bladder tumors, but their goal has been the promise of personalized medicine; specifically, molecular tests to predict outcome and response to intravesical, neoadjuvant and adjuvant/salvage treatments and to guide targeted treatment selection [7,20,28,59,60,61,62,63,64,65,66,67,68,69].

## 5. Stem Cell Origin of BC

First documented more than 30 years ago in hematologic malignancies, cancer stem cells have been described and characterized in many solid tumors including BC, using either clinical specimens, experimental models, or both [28,63,70,71,72]. Progenitor/stem cells are an integral part of normal tissue differentiation and remodeling, as they self-renew and differentiate into various cell types. Similarly, cancer stem cells can explain the clonal origin of cancer with plausible cancer-initiating driver mutations. Furthermore, stem cells can explain how and why lineage plasticity and tumor heterogeneity arise by accumulating genetic alterations (mutations) and from the interactions of cancer stem cells and their hyper-proliferative cell progeny, with the tumor microenvironment [28,63,64,70].

Normal urothelium consists of three cell layers, the umbrella (the upper most layer), the intermediate, and the basal cell layer (Figure 3). In the literature, the turnover of the normal adult urothelial umbrella cells is quoted as anywhere between 3 to 6 months, 200 days, 40 weeks, or even up to a year; indicating that the turnover of unstimulated urothelium is slow [28,73,74,75]. Lineage tracing, lineage depletion, inducible Cre-recombinase model systems, and carcinogen or inflammation-induced mouse models have identified both normal urothelial and BC stem cells.

Two competing models were proposed to explain the generation of three urothelial cell layers. The linear model of urothelial formation and regeneration proposed that the urothelial stem/progenitor cells residing in the basal cell layer (cytokeratin (KRT) 5+, 14+) are capable of self-renewal and give rise to intermediate cells. The intermediate cells, in turn, generate the umbrella (superficial) cells. The competing model is the non-linear model, which proposes that adult urothelial stem/progenitor cells reside in the basal and intermediate cell layers and that both layers give rise to the umbrella cells; a caveat being whether the basal stem cells go through an intermediate cell type to differentiate into the umbrella cells [76,77,78,79]. It is now generally accepted that the origin of urothelial differentiation resides within stem cells that are present in the basal layer. Urothelial injury with subsequent growth stimulatory signals from stromal cells (fibroblasts residing within the lamina propria) activate these stem cells, generating proliferating urothelial cells that differentiate into intermediate cells with umbrella cells representing terminal differentiation. 

## 6. Genetic Hallmarks of Papillary and Non-Papillary Invasive Pathways of BC Development

The two-track model explains the divergent pathways of the development of papillary and non-papillary invasive tumors. This has been further supported by lineage tracing experiments in mouse models [78,79,80,81]. These experiments demonstrate that stem cells that give rise to low-grade tumors (almost never MIBC) and MIBC (almost always high-grade) originate in different cell layers, with histologic variants adding to the complexity. Most low-grade tumors are papillary and arise due to genetic alterations in the urothelium that promote a hyperproliferative phenotype (hyperplasia). The origin of low-grade tumors is traced to the intermediate cell layer [79,80,81]. Contrarily, high-grade NMIBC arises from a different cellular and molecular pathway with early hyperplasia combined with severe dysplasia. In the two-pathway model, Tis was proposed as arising from dysplasia alone. Both high-grade NMIBC and Tis have a high propensity to invade and metastasize [28,79,80,81,82,83]. The two divergent pathways were also proposed to be interrelated, as both high-grade NMIBC and Tis have the potential to progress to MIBC [5,7]. MIBC originates from KRT5+ stem cells within the basal layer with Tis as its precursor. Differences in the genetic landscape of low-grade papillary tumors and of high-grade tumors that begin as NMIBC provide molecular evidence for their different histologic appearance and clinical behavior [84].

In addition to de novo genetic alterations, the interaction of actively proliferating tumor cells with adjacent stromal cells and exposure to varying degrees of hypoxia and altered metabolism (e.g., aerobic glycolysis) contribute to and drive a dynamic genetic landscape that promotes BC (especially MIBC) progression and resistance to treatment [85]. Additionally, variant histologies found in bladder tumors harbor genetic alterations that enhance tumor heterogeneity and the complexities associated with the clinical management of BC [86].

Starting with the studies conducted more than two decades ago, activating FGFR3 mutations, inactivating (loss-of-function) mutations in TP53, RB1, and PTEN genes, as well as copy number alterations in other genes, have molecularly defined the papillary and dysplasia/Tis pathways. Other genetic alterations such as mutations in RAS and Phosphoinositide 3-kinase (especially the catalytic 110α subunit encoded by PI3KCA), MDM2, and CDKN2A genes further support the divergence of molecular pathways that drive the development of papillary and non-papillary tumors [7,28,65,86]. LOH on chromosome 9 (9p, 51%; 9q, 57%; even up to 70%) is both an early, and the most common genetic alteration identified in BC, and it is observed in all stages and grades [87,88,89,90,91,92,93]. The loss-of-function genetic alterations (mutations and loss of heterozygosity) in tumor suppressor genes mapping within chromosome 9p (CDKN2A/p16^INK4a^/p14ARF; CDNKN2B/MTS-2)/p15^INK4b^) and 9q (PTCH1, TSC1, and DBC1) loci were discovered to be common in NMIBC. Contrarily, mutations in the TP53 gene on chromosome 17 were found to favor the development of MIBC [94,95].

Cytogenetic studies and mapping of the molecular alterations have supported the divergent pathway model but have also supported molecular instability associated with disease progression. Cytogenetic alteration analysis showed a “temporal order” in the appearance of chromosomal imbalances with an increase in tumor grade and stage [96]. Molecularly it was proposed that BC arises from two cytogenetic pathways. One of these proposed pathways is initiated by chromosome 9 alterations followed by alterations in chromosomes 11 (−11p) and 1 (1q+). This pathway correlated with stage Ta-T2 tumors. The other pathway was proposed to be initiated by a gain of chromosome 7 (+7) followed by alterations in chromosome 8 loci (8p-; +8q) [7,96]. The evolving state-of-the-art molecular and profiling techniques have mapped the genomic atlas of BC.

FGFR3 activating mutations are common in the papillary pathway. These mutations were found to be independent events in Ta tumors (mostly low-grade); up to 72% of low-grade tumors have mutated FGFR3 [84,97]. Similarly, oncogenic RAS mutations are a hallmark of low-grade papillary tumors; however, FGFR3 and RAS mutations were shown to be mutually exclusive [98,99]. FGFR3 and RAS mutations are associated with a good prognosis and low recurrence rate. Both FGFR3 and RAS signal through the same receptor-tyrosine kinase (RTK)-mediated pathway, where FGFR3 is one of the RTKs that activates the Ras protein. Ras is a small GTP-binding protein that activates the downstream Raf/MAPK pathway. Since FGFR3 is upstream of Ras, a tumor with mutated Ras would not need FGFR3 to drive Raf/MAPK mitotic signaling to induce a hyperproliferative phenotype. The papillary pathway is also characterized by mutations in PI3KCA which activates the PI3K/AKT/mTOR pathway. LOH at TSC1 that results in loss-of-function also activates mTOR signaling causing hyperproliferation [7,65,84]. Additional alterations have been documented in STAG2 (stromal antigen 2; regulates the separation of sister chromatids during cell division), and in KDM6A (lysine-specific histone demethylase) genes. Both STAG2 and KDM2 act as tumor suppressors [84,100]. Mutations in both genes show gender bias and are twice more common in women with low-grade NMIBC than in men [84,100,101].

Starting in the 1990s, single and multi-institutional studies have documented the genomic instability in MIBC. Strategies such as whole-organ genome mapping were/are employed to identify driver mutations in “forerunner genes” and provide evidence for the clonal origin of tumors [19,39,102]. Czerniak’s group proposed the concept of forerunner genes fifteen years ago. This concept was based on whole-organ histologic and genetic mapping (WOHGM) studies for identifying clonal hits associated with growth advantage, thus tracking the development of human bladder cancer from occult in situ lesions [20,39,103]. The initial study identified three “major waves” of genetic alterations that reflect the evolution of normal urothelial cells into cancer cells by possibly acquiring a growth advantage resulting from these genetic alterations. These changes were shown to map to six regions at 3q22–q24, 5q22–q31, 9q21–q22, 10q26, 13q14, and 17p13, and were proposed to be “critical hits” that drive BC development [19]. Among these, chromosome 13q34 locus encompassing RB1 was associated with Tis [20]. MTAP gene on chromosome 9p21 could be another forerunner gene, as its homozygous loss is associated with poor survival [104]. However, since LOH at chromosome 9 is an early event in the development of BC, and the locus also harbors p16 (CDKN2), it emphasizes the need for functional studies to define if a genetic alteration is genuinely a forerunner gene that drives BC development or if it is a bystander [104,105]. The concept of the forerunner gene was and still is an attractive concept. It could explain an initiator driver genetic alteration(s) that leads to the development of low- and high-grade BC, malignant progression of MIBC, treatment resistance and synthetic lethality. This concept possibly gave way to molecular subtypes using bulk “-omic” approaches. However, the bulk -omic approaches are designed to reveal associations between any alteration (genes, transcripts, proteins, or metabolites) and BC (especially MIBC) and not necessarily the drivers of cancer development and progression, as the forerunner gene concept proposed. Therefore, functional and mechanistic investigations beyond the correlative studies are needed to identify the true “drivers” or forerunner—genes, transcript variants, proteins and/or metabolites—as novel targets for developing precision treatments or prognostic markers for patients with advanced BC.

A time-tested seminal conclusion of the early studies which has stood the test time was that tumor suppressor TP53 is the most mutated gene in MIBC. About 50% of MIBC show loss-of-function mutations in TP53 [106,107,108]. Mutated p53 protein is stable and accumulates in the nuclei, but it is non-functional. MDM2, a negative regulator of p53, is mutated or shows polymorphism in about 30% of MIBC. Overexpression and activation of MDM2 removes the “p53-brake” on cell cycle progression and abrogates p53-induced apoptosis of damaged/mutated cells. MDM2 gene amplification has been reported in BC [109,110]. Although MDM2 mutations are less common in BC, the Cancer Genome Atlas (TCGA) mutational analyses studies showed that mutations in TP53 and MDM2 genes are mutually exclusive [111,112,113]. Similar to TP53 mutations, the loss of RB1 is prevalent in invasive bladder tumors. Rb is a regulator of cell cycle progression. Unsurprisingly, bladder tumors harboring the loss of both TP53 and RB1 were associated with a worse prognosis [114,115,116]. However, in mouse models, the concurrent loss of both p53 and Rb is necessary but insufficient to initiate tumorigenesis along the non-papillary invasive pathway [117]. Moreover, in a Phase III clinical study, p53 status was not predictive of recurrence or response to MVAC, in an adjuvant setting [118]. Along with p53 and Rb mutations, PTEN deletions are also found to be common in the non-papillary invasive pathway. PTEN is a phosphatase for lipids, mainly phosphatidylinositol (3,4,5)-trisphosphate (PIP_3_) that triggers PI3K/AKT/mTOR signaling. This is a major signaling pathway that promotes cell survival and proliferation. While these markers are known functional drivers of tumor growth and progression, in recent years, advances in genome-wide sequencing (The Human Genome Project; TCGA datasets) have led to an “explosion” of predictive biomarkers based simply on expression analysis.

## 7. Transcriptome Profiling and the Quest for Personalized Medicine

Within the last decade, exponential advancement in genome-wide profiling of bladder tumors and adjacent normal urothelium and in some cases, whole-organ genome profiling, have significantly improved our understanding of the biology of BC. Some concepts, such as forerunner genes, were introduced to describe genetic events that commit urothelial cells to the papillary or non-papillary (flat) tumor pathways early in development [19,20,60]. The discovery of cancer stem cells, and their transcriptome profiles resembling that of the uro-basal compartment, linked BC to the concept of molecular subtypes that was first introduced in breast cancer [26,119,120,121,122,123,124,125]. Transcriptome profiling and unsupervised clustering identified luminal and basal subtypes in NMIBC and MIBC similar to those observed in breast cancer [61,113,126,127,128,129,130,131,132]. As in the case of breast cancer, the impetus for molecular subtyping of bladder tumors, especially of MIBC, is based on the promise of personalized medicine. Toward this goal, the focus has been on investigating the subtypes’ potential to risk-stratify patients with MIBC regarding the clinical outcome and to guide clinical decisions regarding neoadjuvant treatment decisions [66,69,124,126,127,130,133,134]. In the case of breast cancer, despite more than a decade of research, molecular subtypes have not been reduced to clinical practice. Instead, functional biomarkers (estrogen receptor, progesterone receptor, HER-2 status, etc.) are routinely used to guide treatment decisions for an individual patient [26,135,136].

Starting with early molecular taxonomy initiatives, such as the Lund-Taxonomy and TCGA clusters, to the most recent Consensus Molecular Classification, at the highest level, bladder tumors have been divided into two major subtypes: “basal” and “luminal” [61,113,129,130]. The basal subtypes share an expression profile with the basal cells of the urothelium. The luminal subtypes share a transcriptome profile with the differentiated urothelium [26,129]. The basal subtypes have been reported to associate with poor prognosis, whereas the luminal lineage tumors associate with a better prognosis [126,127,137,138]. Along with an emphasis on predicting clinical outcomes, many studies have focused on MIBC subtypes. However, the original Lund-Taxonomy classification from 2012 showed an association of the subtypes with tumor grade. In this classification, the best cancer-specific survival was associated with the “UroA” subtype (now “Urothelial-like”), which represented low-grade NMIBC and had a molecular signature similar to the LU subtype [129]. Low-grade tumors are of luminal lineage, almost always are NMIBC, and inherently have a good prognosis [139,140]. Intra-tumor co-existence of basal and luminal tumor regions has also been reported in MIBC patients, as MIBC subtypes exist in NMIBC [26,141,142,143].

Over the years, molecular classifications of BC, particularly MIBC, have evolved, leading to subdividing of luminal and basal lineage tumors into more subtypes. The Lund-Taxonomy has divided luminal lineage tumors into urothelial-like (Uro or Uro-like) and genomically unstable (GU). The basal lineage is sub-divided into basal/squamous, mesenchymal-like, and small cell/neuroendocrine-like (Figure 4). The TCGA Research Network has had at least three molecular taxonomy classifications—TCGA clusters, TCGA-2017, and the TCGA Consensus [51,65]. The Consensus classification has subtyped MIBC into six categories—luminal papillary, luminal non-specified, luminal unstable, stroma-rich, basal/squamous, and neuroendocrine-like (Figure 4) [61]. Furthermore, multilayer heterogeneity has been reported within the basal/squamous subtype [144]. These classification systems reflect the intrinsic heterogeneity among bladder tumors, which may start at their cellular origin, but become more complex as the tumors progress and are further influenced by the host cellular *milieu*, tumor-stromal interactions, stress (e.g., hypoxia), and treatment. This would suggest that inherently, the molecular subtype of a tumor will be transient.

Although in transcriptome profiling, studies subtyping is performed by unsupervised hierarchical clustering of thousands of transcripts, among most studies, only a few transcripts are common [69,126,127,129,130,137,138,142,145]. In most studies, basal lineage tumors express basal cytokeratins (KRTs)—5, 6 (A—C), and 14. Additional genes reported in two or more studies for basal lineage MIBC are CD44 (a stem cell marker), epidermal growth factor receptor (EGFR), CDH3, interleukin-6, TWIST, SNAIL, vimentin, and ZEB2. In most studies, the common transcripts for luminal lineage tumors are KRT20, GATA3, uroplakins (UPK) 1B, 2, and 3A, and FOXA1. Some additional transcripts included in two or more studies are CLDN3, CLDN4, CLDN7, ERBB2, PPARG, FGFR3, CYP4B1, and XBP1 [26,69,126,129,130,133,138,142,145].

The majority, if not all, of molecular subtyping classifications have been performed by bulk tumor transcriptome profiling. Such methodologies have been proposed for predicting the “kinetics” of tumor progression and patient survival. More importantly, transcriptome profiling of primary tumors is proposed for individualizing a patient’s treatment options [51,67,69,83,127,130,133,137,138,142,145]. However, a tumor, especially an aggressive tumor such as MIBC, is not “static”. It is continuously responding to and metamorphosing in the temporal and spatial context. In fact, such multiple cell lineages offer a heterogeneous tumor the ability to grow, escape immune surveillance, become resistant to treatment (innate or acquired), and to metastasize. Metastasis, which is an insufficient process, would require that certain cell lineages in a tumor either have or acquire an ability to extravasate, survive in circulation under varying degrees of oxidative stress, and then to intravasate for colonizing a distant organ. The different cell lineages within a bladder tumor likely dictate the differential response to therapy. This is potentially due to the accumulation of different genetic alterations and transcriptome profiles which subsequently dictate functional pathways that drive resistance to therapy. A recent study demonstrated that immune cell infiltration and inflammation in the pretreatment tumor microenvironment assessed by a novel gene signature and not molecular subtypes were associated with response to BCG in NMIBC [146]. Intriguingly, FGFR3 mutations and overexpression were both associated with low immune signatures and the lower response rate for perioperative platinum-based chemotherapy [146,147]. A negative immune checkpoint signature (PDL-1 expression) is also potentially associated with worse clinical outcome for patients with squamous cell carcinoma of the bladder [148].

Although pathologists have documented tumor heterogeneity and tumors with mixed histology, single-cell sequencing of bladder tumors has provided the necessary proof for lineage plasticity [67,86,149,150]. Single-cell sequencing has identified that a single cancer cell can simultaneously express multiple subtypes [67]. Consistent with the evidence of tumor cell lineage plasticity and inherent tumor heterogeneity, immunohistochemistry analyses have shown that within the same high power field, adjacent tumor cells within a single MIBC specimen express different levels of a single marker, ranging from high to no expression [142]. These results show that categorizing bladder tumors into specific molecular subtypes by bulk transcriptome or any other “-omic” profiling may reveal the status of that tumor and its microenvironment (stromal, immune cells) at one time point and in one portion of the tumor, from where the specimen was obtained for profiling. Furthermore, a single tumor could be assigned different subtypes based on how many cells of a particular lineage are present in the sample and the level of expression of the markers used for subtyping in the temporal and spatial context. This may not necessarily translate into a tumor’s behavior; a few cells could drive a tumor’s lethality, but the profiles of such cells may be missed in bulk transcriptome profiling. For this reason, recent studies have reported the subtypes’ inability to predict clinical outcome [67,142,151].

By comparing multiple classification systems (MCG-1, Lund-Taxonomy, TCGA-2017, and the Consensus; Figure 4), the authors (DSM, VBL) recently reported that molecular subtypes by each of these classifications strongly correlate with tumor grade.

The TCGA BC dataset with 402 MIBC specimens has been used in most studies for MIBC subtyping and for predicting clinical outcome. Curiously, this dataset includes twenty-one low-grade tumors as MIBC. However, MIBC is almost never low-grade, and the removal of these low-grade tumors from this TCGA dataset abrogates the ability of the subtypes to predict overall survival [132,142]. Similarly, using the UROMOL NMIBC dataset of 476 specimens, subtypes have been reported to predict clinical outcome and disease progression [132,139,152]. In this dataset, >50% of the specimens are low-grade, only thirty-one of the NMIBC patients have progressed to MIBC, and twenty-two of them are high-grade tumors [139]. This shows that even in this dataset, the subtypes’ ability to predict clinical outcome is primarily because of their ability to identify tumor grade. It is well-established that given sufficient time, high-grade tumors will progress to MIBC. The strong association of MIBC subtypes with tumor grade has completed a full circle and returned to the original divergent pathway model for developing low- and high-grade tumors that Droller proposed some forty years ago [5,6,8]. Nevertheless, based on molecular heterogeneity among BC and its association with BC progression, it would be important to study how the intermediate grade BC (grade 2 per 1973 WHO classification) and intermediate risk NMIBCs fit into the two-pathway model. The implementation of the 2004 WHO/ISUP system has been reported to either underdiagnose high-grade BC or over-diagnose BC as high-grade [153,154,155]. In this regard, molecular subtyping could be useful in accurately determining bladder tumor grade. This would have immediate clinical significance, as the management of patients with low- and high-grade tumors is vastly different.

Profiling based on immunohistochemistry has been proposed considering the occurrence of intra-tumor heterogeneity and lineage plasticity [26]. Grouping tumors based on the immunohistochemistry algorithm may allow the inherent heterogeneity in terms of marker expression at the single cell level to be taken into consideration. Another supporting concept would be to exploit tumor heterogeneity to predict clinical outcome. However, this would have to be performed on a “whole tumor” basis to avoid missing out on “clinically significant but small foci” within a tumor. Taken together, although molecular subtyping in the present form may not be ready for clinical practice, it has helped to crystalize the significance of intra-tumor heterogeneity and lineage plasticity as the next frontier to improve personalized care for BC patients. Further, as has been realized for breast cancer, functional drivers and not molecular subtypes assigned by transcriptome profiling per se may be more clinically useful in predicting a bladder tumor’s clinical behavior [156,157,158,159]. One could develop an algorithm based on such functional drivers to predict patients’ clinical outcome and individualize their treatment options.

## 8. Genomic Alterations/Tumor Mutation Burden in BC

It is well-known that every tumor harbors DNA alterations/mutations. One of the first comprehensive attempts to catalog a consensus of cancer genes showed that more than 1% of the genes in the human genome contribute to cancer [160]. Cataloging of cancer mutations was initiated when 4,938,362 mutations from 7042 cancers were documented; the study described more than 20 distinct mutational signatures. The study described that in addition to the genome-wide mutational signatures, hypermutations localized in small genomic regions, “Kataegis”, are found in many cancers [161]. COSMIC, the Catalogue Of Somatic Mutations In Cancer (https://cancer.sanger.ac.uk, accessed on 13 April 2022), is a widely used and heavily cited comprehensive resource for exploring the effect of somatic mutations in human cancer [162]. The most recent COSMIC v86 version includes nearly 6 million mutations in coding regions across 1.4 million tumor specimens, which are curated from over 26,000 publications. COSMIC includes all somatic mutations that promote cancer growth and progression and includes non-coding mutations, gene fusions, copy number variants, and drug-resistance mutations [162]. However, COSMIC is a “live document” as novel mutations are found across all chromosomes in various cancer types and tumor specimens in a random fashion. The random nature of the mutations across the entire human genome (i.e., no physical association) indicates that although there are “driver mutations”, the majority of the “coordinated regulation” may be at the transcriptional and epigenetic levels [163].

Studies on the TCGA dataset have revealed that BC has a higher mutation rate exceeded probably only by lung cancer and melanoma [164,165]. About 65% of mutations found in cancers are mediated by APOBEC (apolipoprotein B mRNA-editing enzyme, catalytic polypeptide-like) family of cytidine deaminases. APOECs affect many signaling pathways involved in cancer growth and progression. Mutations in chromatin-modifying genes and transcription factors are also common in BC [164]. The human genome contains several homologous APOBECs; APOBEC1 and APOBEC2 account for most of the point mutations [164]. Cytidine deaminases convert cytosine to uracil and, similar to the most studied “activation-induced cytidine deaminase”, the APOBEC class of enzymes is involved in both innate immunity and RNA editing [166]. Upregulation of APOBEC(s) in tumors enhances C→T transitions and C→G transversions with preference to 5′TCW (W=A/T) or TCG DNA trinucleotides. Genomic deamination events lead to cytosine mutagenesis clusters (kataegis) causing chromosomal aberrations such as translocations [165,166,167,168]. These simultaneous mutation clusters found within the same strand are generated by base alkylation of long single-strand DNA, which forms at double-strand DNA breaks/chromosomal arrangement break points and replication forks. Therefore, APBOEC family induces hypermutations by causing multiple simultaneous changes in randomly created ssDNA [169]. In BC, a mutational pattern is altered based on APOBEC enrichment. APOBEC-high tumors have a significantly higher frequency of mutations in the TP53/RB1 pathway, DNA damage response, and chromatin-modifying genes, whereas APOBEC-low tumors show a higher frequency of KRAS and FGFR3 mutations [170]. In the TCGA-2014 BC dataset with 131 urothelial carcinoma specimens (all high-grade MIBC), 51% showed Tp*C- > (T/G) mutations and high expression of APOBEC3B [113]. An independent report confirmed the upregulation of APOBEC3 activity in BC [171]. In the context of 238 TCGA BC specimens, two variations of APOBEC mutation signatures were found. One consists of C→T transitions (TCW motif) accounting for 17% of single nucleotide variations, and another accounting for 48% of single nucleotide variations (both C→T and C→G mutations) [164]. In the TCGA-2017 dataset results on 412 specimens were reported [130]. In that study APOBEC mutation signature was observed in 67% of the tumors and mutagenesis correlated with the expression of APOBEC-3A and APOBEC-3B [130]. Most of these mutations had a clonal origin, suggesting that APOBEC-mediated mutations occur early in the development of BC. It is noteworthy that Patients with high APOBEC-signature mutagenesis and high mutation burden showed significantly higher (75%) 5-year survival probability. Better survival was also seen in subsets defined by high mutation burden or high APOBEC-signature mutation load [130]. A study by Glaser et al. included 395 specimens from the TCGA dataset and showed that APOBEC-high tumors have better overall survival than those with APOBEC-low tumors [170].

As some of us have reported earlier, the TCGA-2017 lists 436 specimens. Among these 23 are normal bladder, one “metastasis”, five with missing RNA-Seq data, 5 NMIBC (T0 N0 M0: 1; T1N0 M0: 3; TX: 1), and 402 listed as MIBC. Among the 402 MIBC specimens, 21 are “low-grade” tumors [142]. Therefore, the 412 specimens reported in the TCGA-2017 study and 395 specimens reported in the Glaser et al. study contain both low-grade tumors (which are rarely MIBC), and/or NMIBC specimens [59,130,170]. Since APOBEC mutations likely occur early in the development of BC, if would be clinically significant to reanalyze the data on the 412 or 395 specimens to determine if low-grade and high-grade tumors have an equal frequency of APOBEC-mediated mutations and if the mutations are similar in nature. Furthermore, it remains to be determined if the correlation of APOBEC mutations with better survival also depends on the presence of the low-grade tumors in this dataset. Notably, in the Beijing Genomics Institute (BGI) cohort and in a subset of MIBC specimens, no significant correlation was observed between survival and APOBEC levels [59,170]. In the broader context, such an analysis could reveal further proof of the divergent molecular pathways for the development of low- and high-grade tumors and could further elucidate whether these pathways are interrelated (i.e., similar mutation pattern or dissimilar mutation frequency/pattern) and how they drive the clinical course of BC.

Mutational analysis of 5243 cancer genome sequences (4633 exomes and 610 whole genomes) of smoking-related cancers has revealed a signature (signature 5) that harbors inactivating mutations in ERCC2 which encodes a component of nucleotide excision repair. The signature was found to be associated with BC and was higher among smokers compared to non-smokers [172]. This signature is characterized by T→C and C→T mutations with a transcriptional strand bias for T > C mutations [172]. Analyses of the TCGA dataset for mutations among smokers and non-smokers with MIBC found only one gene that was mutated—GPR15—and it was found to be up-regulated in both luminal- and basal-subtype tumors from smokers [173]. It is noteworthy that in the TCGA dataset, all low-grade tumors (listed as MIBC), are of the luminal subtype. Therefore, finding a smoking-related mutational signature using this dataset may not reveal how smoking changes the mutational landscape during BC development [142]. Tumors with somatic mutations in ERCC2 and other nucleotide excision repair genes (signature 5*) were also found to be associated with a higher mutation burden in urothelial tumors compared to those with wild-type ERCC2 and other nucleotide excision repair genes [59,142]. A composite mutational signature of ERCC2 deficiency was found to correlate with cisplatin response in patients with advanced BC [174]. ERCC2 missense mutations are also enriched in primary MIBC (treatment naïve) compared to secondary MIBC (resulting from NMIBC progression) and is possibly a predictor of increased sensitivity to cisplatin-based therapy in the neoadjuvant setting [175,176].

As reviewed recently by Cooley et al., the TCGA-2017, MSKI, and BGI datasets have reported enrichment of mutations in MIBC in various cellular pathways. Among these, mutation frequency (≥50%) is found in cell cycle-related genes (e.g., TP53, RB1, CDNK1A, CDNK2A), chromatin regulatory genes e.g., (KTM2D, KDM6A, STAG2, ARID1A, KTM2C, EP300, CREBBP, BRWD1), and in RAS/PI3-K pathway genes [59]. A slightly lower (20–30%) but significant mutation frequency has been noted in the chromatin remodeling genes (reviewed in [59]). Whole-organ mapping for studying the locoregional changes has revealed DNA methylation alterations and their association with the initiation of field-effect in BC. The field-effect likely occurs concurrently with sub-clonal low-allele frequency mutations and a small number of DNA copy alterations. The study found an initiating mutation in ACIN1 in the normal mucosa with 21 additional mutations associated with cancer progression [39]. The role of ACIN1 in bladder carcinogenesis remains to be evaluated. The mutational landscape studies have advanced the molecular understanding of the field-effect in bladder carcinogenesis and the pathways involved in BC progression. Multi-omic approaches involving genomics, transcriptomics, epigenomics, and proteomics analyses confirm genomic instability in MIBC and certain gene signatures associated with poor response [177].

Along with the genomic/transcriptomic approaches, metabolomic analyses of urine specimens, the availability of the Human Metabolome Database, and the bulk epigenome analyses approaches have identified altered pathways in BC, some of which are likely associated with bladder carcinogenesis, growth, and progression to MIBC. Recent reviews describe these discoveries [178,179,180,181,182]. Currently the majority of the “omic” studies at all levels (gene, epigenetic, transcript, protein, metabolites) are correlative. These approaches should pave the way for the much-needed systematic, mechanistic, and functional studies for developing personalized, targeted, and synthetic lethality treatment regimens and clinically significant prognostic markers for BC.

## 9. Urine Biomarkers for BC: Clinical Need and Reality

The question for a PSA-like test for BC has driven the quest for finding diagnostic and surveillance biomarkers/tests for BC [183]. In another context, the clinical need for accurate prediction of metastasis, response to treatment in both the neoadjuvant and adjuvant/salvage settings, and survival has led to the discovery of several potential prognostic markers. Searching for biomarker-related articles in PubMed with the search term “Bladder cancer and marker” returns 12,300 entries. Corresponding to the advent of high-throughput assays and “omic” technologies, more than half the number of biomarker-related articles in PubMed were published within the last ten years. Most articles on BC biomarkers describe the discovery and development of a biomarker test. Some biomarkers advance to the validation stage, but mostly in single-center studies; even fewer advance to multi-center studies, and none have been recommended for clinical practice.

The reason for BC being at the front-and-center of biomarker development is the fact that any cancer determinant—genetic or protein-based—whether it is cell-associated (DNA, RNA, cell-surface antigens, intracellular proteins) or soluble (secreted proteins, free circulating DNA, RNA, exosomes, glycosaminoglycans), can be detected in urine or in exfoliated cells. However, the quest for urine biomarkers started with the detection of hematuria for diagnosing BC and other urological cancers in the late 1980s [23,184,185,186]. In these trials, urological cancers were found in <10% of the patients with the percentage of hematuria causing benign diseases three times higher. Each of the false-positive results require a complete cystoscopy workup. Along with the prevalence of BC among patients with gross hematuria being 10–25%, and even lower (2–10%) if it is asymptomatic microscopic hematuria (ASM), screening of the general population for hematuria to detect BC is not practical [187,188,189,190,191,192,193,194,195,196,197,198]. In fact, with the worldwide incidence of BC about 9.6 per 100,000 in men and almost four times lower among women [199], screening of adults over the age of fifty years even with an ideal biomarker will be impractical, due to a significantly high percentage of false-positive cases requiring a complete workup [185,200,201]. However, among patients with a presenting symptom of painless hematuria, where the prevalence of BC is relatively high, a non-invasive biomarker could be useful, provided it has high sensitivity (i.e., high negative predictive value). The goal is that the marker should not miss a tumor, especially high-grade, as these tumors have propensity to become MIBC. Since the standard clinical practice for ASM workup for patients thirty-five years or older is cystoscopy, a marker with sensitivity comparable to cystoscopy and reasonable specificity (i.e., positive predictive value) could, in principle, gain acceptance. However, while the EAU AUA/SUFU guidelines recommend upper-tract imaging depending on a patient’s risk, they recommend against the use of non-invasive biomarkers for the initial evaluation of ASM. Most, if not all, FDA-approved urine tests have also yielded inconsistent results and/or suboptimal efficacy. Although some urine-based biomarkers have shown sensitivity comparable to cystoscopy, they have not been tested in multicenter studies or randomized clinical trials [202,203,204,205,206,207,208,209,210].

A justifiable clinical niche for non-invasive and preferably urine biomarkers is surveillance, as non-MIBC is treated with tumor resection and patients are monitored with cystoscopy at scheduled intervals to detect BC recurrence. For these patients, a marker could be alternated with cystoscopy, to increase the intervals between the procedures. As in the case of painless hematuria workup, the marker should be non-inferior to cystoscopy in detecting the recurrence. The bar for specificity could be “reasonable but not necessarily stringent”, since the patients would have undergone cystoscopy in the absence of the marker anyway. Active surveillance especially for low-grade bladder tumors has been suggested as a “win-win” for both patients and urologists, and the International Bladder Cancer Group has recommended stopping surveillance cystoscopy after five years for patients with “stable, low-grade-appearing recurrence” [211,212,213]. Among patients with low-grade tumors, more than the recommended cystoscopies result in significantly more tumor resections, however, frequent cystoscopy is not associated with time to progression or death due to BC [214]. Since low-grade tumors rarely (almost never) become MIBC, unless there is a high-grade recurrence, patients with stable low-grade recurrence could be followed by an active surveillance protocol that includes monitoring with a non-invasive biomarker that has high sensitivity. However, despite this clinical niche, no marker is used even in the surveillance of patients with a stable low-grade disease [215].

The possibility of false negative results even for low-risk low-grade tumors and false positive results, up to 30% for some markers, is the main reason that no BC biomarkers have been been recommended for use in clinical practice. Although this is similar to the performance of PSA when it comes to diagnosing prostate cancer, BC is held at a higher standard, because cystoscopy, although uncomfortable and invasive, is performed during an office visit, whereas prostate biopsy is not [24,216]. Furthermore, a single BC marker/test may not have the same efficacy across the entire spectrum of BC progression, i.e., from the diagnosis and surveillance for monitoring recurrence, to predicting response, to treatment (BCG or neoadjuvant chemotherapy). Therefore, after the discovery and/or validation of hundreds of BC markers, a question that was posed over thirty years ago about PSA for BC is still valid, but could be revised as follows: “Would a PSA-like test for BC be acceptable even just for reducing the frequency of cystoscopy but never replacing it”? [183,215].

## 10. Ease of Developing Urine Biomarkers

Urine cytology is the standard non-invasive biomarker for BC. However, due to its low sensitivity (especially for low-grade tumors) and inconclusive inference (atypical cells), it is an adjunct to cystoscopy. A recent report showed that multifunctional nanoprobes (Fe_3_O_4_@SiO_2_, mNPs) with positive charges could capture tumor cells from urine and improve the diagnostic efficacy of cytology from about 40% to ~80%, with comparable specificity except in patients with urinary tract infection [217].

Urine is an ideal liquid biopsy material because tumor-associated soluble materials and tumor/urothelial cells are released in the urine when it is stored in the bladder. As a result, virtually any molecule associated with tumor cells or tumor-associated stroma including soluble proteins, enzymes, glycosaminoglycans, cell-surface markers, DNA (mutations, loss of heterozygosity, methylation), mRNAs, non-coding RNAs (microRNAs, lncRNAs), lipids, and metabolites can be measured in urine specimens (Figure 5).

Many of these markers have been recently reviewed in several articles [24,27,218,219,220,221]. Table 1 lists the FDA-approved tests and newer markers that utilize a multiplex format. Along with the advancement of multiplex assays, the current emphasis is on the simultaneous assaying of multiple biomarkers to achieve better diagnostic ability. ONCURIA is a multiplexed bead immunoassay that measures angiogenin (ANG), apolipoprotein E (APOE), alpha-1 antitrypsin (A1AT), carbonic anhydrase 9 (CA9), interleukin-8 (IL-8), matrix metallopeptidase 9 (MMP9), matrix metallopeptidase 10 (MMP10), plasminogen activator inhibitor 1 (PAI1), syndecan 1 (SDC1), and vascular endothelial growth factor (VEGF) levels in urine and creates a validated signature to diagnose BC [209]. In prospective studies, the test has shown 90% or more sensitivity; furthermore, false positive results due to whole blood were shown to be negated by centrifugation of the samples [209,222,223,224].

Several transcript-based tests have been described recently. The two recent transcript-based tests are Cxbladder (Cxb) and XPERT© Bladder Cancer Monitor. The Cxbladder (Cxb) is a combination of three tests with different emphasis. All tests use mRNA biomarkers (IGBP5, HOXA13, MDK, CDK1, and CXCR2) in a multiplex setting and are designed for diagnosing BC and for monitoring its recurrence. Cxbladder Detect (CxbD) is designed for patients who require further workup for BC. Cxbladder Resolve (CxbR) was developed for use after both CxbT and CxbD tests, to accurately segregate test-positive patients likely to have high-impact tumors. The test could detect BC in cases with atypical cytology and hematuria [206,225,226,227,228].

The XPERT© Bladder Cancer Monitor, (XBCM) test measures the levels of five target transcripts (ABL1, CRH, IGF2, UPK1B, and ANXA10) by RT-qPCR. The results of this test (or “linear discriminant analysis”) depend on a regression algorithm that utilizes the cycle threshold results of the five mRNA targets. The test has lower sensitivity for low-grade tumors and missed about 25% of the high-grade recurrence. Nevertheless, XPERT test with two negative tests avoids nearly 75% of unnecessary cystoscopies [229,230,231,232,233,234]. ADXBLADDER™ test measures the expression of MCM5 transcript [235,236,237]. In a large blinded prospective study ADXBLADDER™ test had an overall sensitivity of 44% but was able to exclude the presence of the most aggressive tumors with a negative predictive value of 99%. In another study, the test’s sensitivity was lower than urine cytology [235,237].

Several mutation/DNA methylation-based tests have been recently described and include Assure MDx, UroSEEK, Uromonitor, and EpiCheck. The Assure MDx detects mutations in FGFR3, TERT, and HRAS, and methylation of OTX1, ONECUT2, and TWIST1. The test has not been extensively studied [210,238,239]. The UroSEEK assay, which does not appear to have been commercially developed, incorporates massive parallel sequencing assays for mutations in 11 genes and copy number changes on 39 chromosome arms. The assay is designed to detect alterations in 11 genes. It is performed as a single-plex PCR assay for the evaluation of the TERT promoter region (TERTSeqS), followed by a multiplex PCR assay to amplify the regions of interest in 10 genes (FGFR3, PIK3CA, TP53, HRAS, KRAS, ERBB2, CDKN2A, MET, MLL, and VHL; UroSeqS) [240,241]. The assay may have high efficacy and early detection capability but has not been widely tested [242,243]. The Uromonitor test evaluates hotspot mutations in TERT and FGFR3 genes. The Uromonito-V2^®^, U-monitor evaluates KRAS mutations in addition to TERT and FGFR3 by qPCR [244,245]. Bladder EpiChekck (BE) test is a DNA-based urine assay that evaluates the methylation profile of 15 genomic loci which are frequently methylated in BC cells. The efficacy of this test shows large variation, however a recent study with limited patients (only 40) showed higher accuracy [246,247,248,249,250].

Four tests (BTA Stat/BTA TRAK, NMP22, UroVysion and uCyt+) are FDA approved for BC diagnosis and follow-up (Table 1; [200,207,208,251,252,253,254,255,256,257,258]). Compared to the other tests, uCyt+ was shown to have better sensitivity for low-grade BC, whereas UroVysion was shown to detect BC recurrence before clinical detection [200,207,251,255]. Some newer markers have shown potential to reduce the frequency of cystoscopy up to 70%, however, they have not been extensively tested [206,224,226,227,234,259,260,261,262]. Furthermore, no urine biomarker is recommended to use in clinical practice due anxiety/bias towards missing a tumor, low sensitivity of several markers to detect low-grade tumors, and lower specificity for benign conditions, especially those presenting with hematuria and additional cost [187,188,189,190,191,192,193,194,195,196,197,198,263].

## 11. Clinical Translation of Urine Biomarkers: Where Are We?

One of the impediments to BC biomarkers gaining acceptance in clinical practice is non-adherence to uniform criteria for the evaluation of a biomarker’s performance. Some impediments are the complexity of some of the assays, differences in study populations, inconsistencies in assay methodologies (including sample collection and preservation), and inconsistent cut-off limits to evaluate a marker’s performance, which are contributing factors to the variation in efficacy for the same marker in different studies (Table 1). Meta-analyses studies report the mean and median efficacy parameters for a urine test; however, the results of such analyses hinge on the types of studies included in the analyses. For example, in some studies, patient cohorts may be from an academic/tertiary care center, which usually has a high prevalence of MIBC cases and/or large tumors requiring complex multi-modal treatments. In contrast, others may include both high- and low-risk BC cases. Many markers tend to have high sensitivity to detect MIBC and large tumors. In terms of the control population some studies may include a large percentage of patients under surveillance or healthy individuals, while others may include patients presenting with hematuria and/or benign urological conditions. The confounding factors that could affect a marker’s performance include age, sex, race/ethnicity, co-morbidities, and prior treatments. Such differences in the study populations yield significant differences in a marker’s reported efficacy parameters (i.e., sensitivity, specificity, positive and negative predictive values) of a marker in different studies (Table 1). Therefore, unless a meta-analysis accounts/controls for the different variables across the studies, the reported mean and median efficacies would not be reliable, affecting a marker’s clinical translation [200,251,264].

To counter some of the biases and inaccuracies associated with biomarker studies, the STARD (Standards for Reporting of Diagnostic Accuracy Studies), PRISMA (The Preferred Reporting Items for Systematic reviews and Meta-Analyses), and the updated PRISM-DAT (PRISMA-Diagnostic Accuracy Test) statements/guidelines could be adopted in the meta-analyses and systematic reviews of BC biomarkers [264,265,266,267,268]. Nevertheless, very few biomarker studies for malignant or benign diseases adhere to these guidelines. Attempts have been made to construct a clinical trial design for a head-to-head comparison of a marker-based approach with the standard work-up involving cystoscopy [192]. One of the suggested protocols is a comparison of the standard work-up versus including a biomarker in the standard-of-care surveillance protocol based on cystoscopy; especially for patients with low-risk BC, where active surveillance has been suggested [211,212]. While a fully randomized design that monitors patients by a marker or standard work-up would be ideal [192], considering the reluctance of urologists and patients to adopt a biomarker even for low-risk disease, the possibility that such a clinical trial will be conducted is low. From reviewing the biomarker studies, although urine is a perfect liquid biopsy and BC is at the forefront of biomarker development, inclusion of a biomarker in clinical practice will require that such a biomarker is far superior to PSA, and at least non-inferior to cystoscopy.

## 12. Conclusions

The last forty years have witnessed significant discoveries regarding the molecular oncology of BC. Some concepts, such as the divergent model of BC origin first proposed by Droller over forty years ago, have withstood the test of time. Others, such as the molecular subtypes, have enabled a better understanding of lineage plasticity and tumor heterogeneity but have also revealed the challenges/deficiencies to implementing their use in personalized patient care. The research has revealed the need for discovering functional drivers as markers and targets for therapy to ultimately achieve the goal of personalized medicine, where the bulk- or even single-cell “-omic” approaches may fall short due to tumor heterogeneity. Using the current genomic, proteomic, and cellular technology, several BC markers have been discovered and are being validated. These biomarkers show significantly higher efficacy than what PSA shows for prostate cancer. However, no BC marker has been used in clinical practice. Therefore, Soloway’s first question over thirty years ago regarding PSA for BC is still relevant today. Along with tremendous progress in clinical–translational research related to the molecular oncology of BC, personalized treatment for patients with BC could be a routine clinical practice soon.

## Figures and Tables

**Figure 1 cancers-14-02578-f001:**
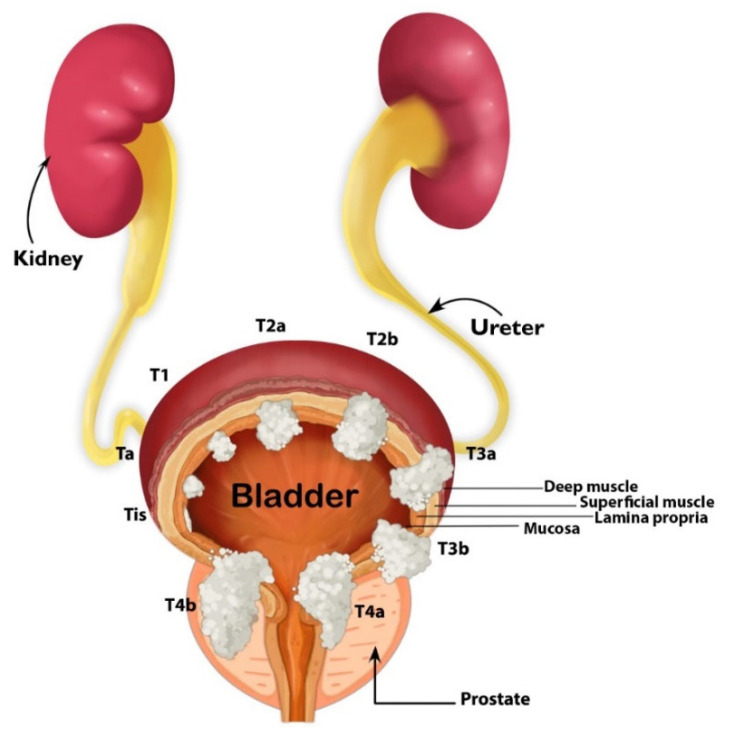
Schematic representation of the genitourinary system with bladder tumor stages. Illustration by Maite Lopez.

**Figure 2 cancers-14-02578-f002:**
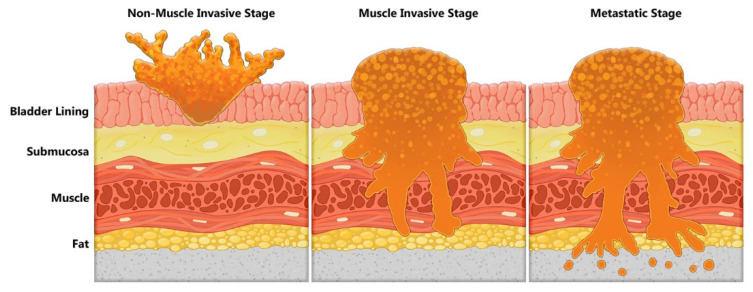
A schematic representation of urothelial tumors. The invasion of a bladder tumor through various layers are shown. Low-grade tumors remain confined to the urothelial layer, whereas high-grade tumors will invade through submucosa, detrusor muscle, and beyond. Illustration by Maite Lopez.

**Figure 3 cancers-14-02578-f003:**
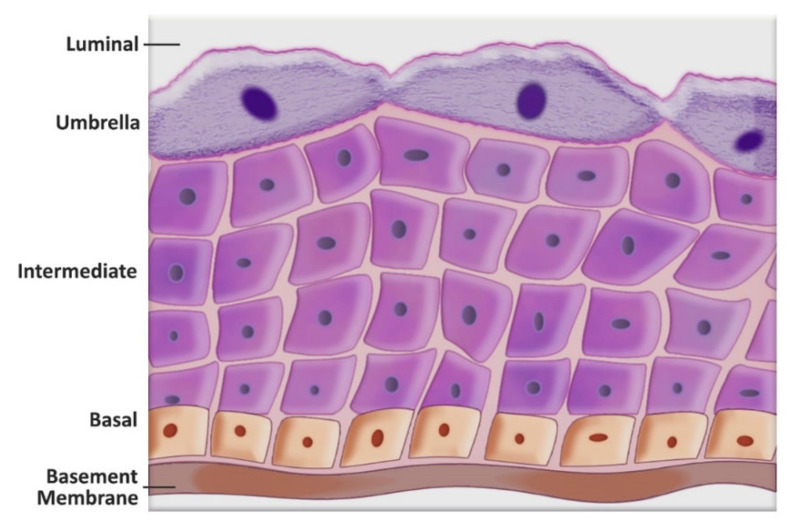
A schematic representation of the bladder urothelial layers. Different cell layers likely give rise to low- and high-grade bladder tumors. Low-grade tumors likely originate from the intermediate cell layer. Muscle-invasive bladder cancer (almost always high-grade) originates from KRT5+ stem cells within the basal layer with Tis as its precursor. Illustration by Maite Lopez.

**Figure 4 cancers-14-02578-f004:**
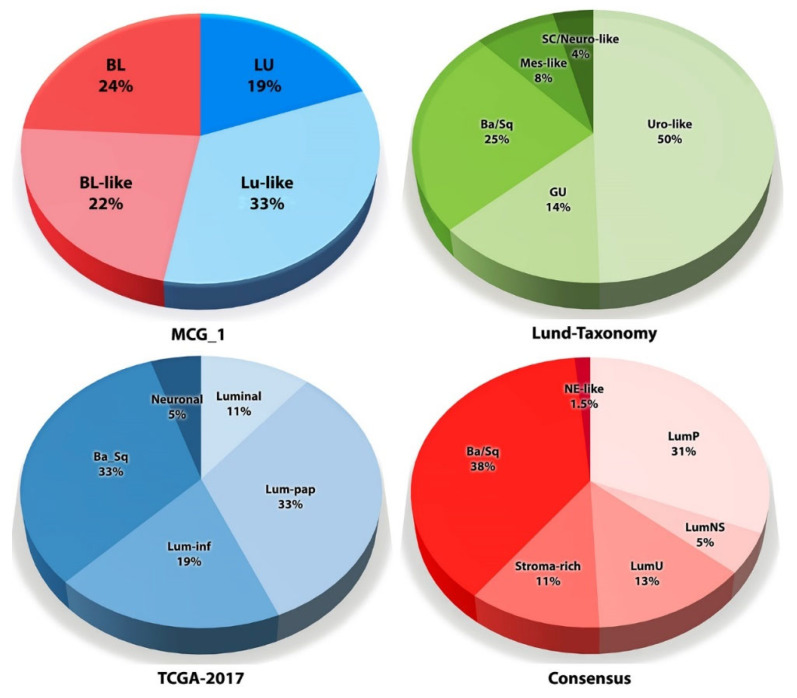
Muscle-invasive bladder cancer molecular subtypes by different classifications. Subtype distribution of the MIBC specimens in the TCGA bladder cancer dataset based on published classification systems (ref. [142]). Illustration by Maite Lopez. BL: basal; Lu: luminal; LumP: luminal papillary; LumNS: luminal non-specified; NE-like: neuroendocrine-like; Lum-pap: luminal-papillary; Lum-inf: luminal-infiltrated; Ba/Sq: basal/squamous; Uro-like: urothelial-like; GU: genomically unstable; Mes-like: mesenchymal-like; SC/Neuro-like: small cell/neuroendocrine-like. Illustration by Maite Lopez.

**Figure 5 cancers-14-02578-f005:**
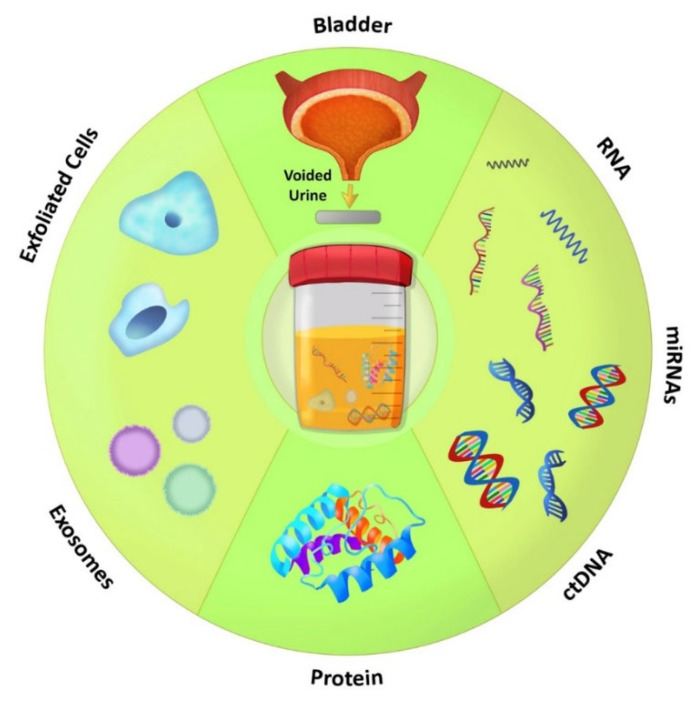
Urine-based biomarkers. A schematic representation of the various materials in urine specimens that have been used for the development of urine biomarkers. Illustration by Maite Lopez.

**Table 1 cancers-14-02578-t001:** Urine biomarkers: List of current FDA-approved + commercialized urine biomarkers.

Test/Manufacturer	Molecule Detected and Assay Type	Reported Sensitivity%	Reported Overall Specificity	Reference
BTA Stat^®^ Polymedco™	Complement factor H and related protein. Soluble marker POC Approved for BC diagnosis and surveillance	64 (58–69); 58 (29–74) Modest: low-grade and low- volume tumors	77 (73–81) 73 (56–86) Low: benign urologic diseases; hematuria	[155,162,163,206]
BTA TRAK^®^ Polymedco™	Same as above Soluble marker Quantitative test (ELISA)	65 (54–75)	74 (64–82) Could be ~50% for benign conditions	[155,162,163,206]
ALERE NMP22^®^ TEST Abbott	Nuclear mitotic apparatus protein Soluble marker quantitative test (ELISA) Approved for diagnosis in high-risk individuals and surveillance.	69 (62–75) 65 (50–80)	77 (70–83) 81 (40–87) Modest for some benign urologic diseases	[155,206,207]
ALERE NMP22^®^ *BladderChek*^®^ Test Abbott	Same as above Soluble marker POC test	58 (39–75)	88 (78–94) Modest for some benign urologic diseases	[155,206,207]
Vysis^®^ UroVysion diagnostic test Abbott, Vysis	FISH test to detect alterations in chromosomes 3, 7, 17, and 9p21 Exfoliated cells Diagnosis and surveillance	63 (50–75) 72 (69–75) 76 (65–84)	87 (79–93) 83 (82–85) 85 (78–92)	[155,162,163,206,207,210]
ImmunoCyt/ uCyt+™ Scimedex	CEA and two bladder tumor cell-associated mucins Immunocytochemistry Exfoliated cells Approved for surveillance	81 (42–100) 84 (77–91) 78 (68–85)	75 (62–95) 75 (68–83) 78 (72–82)	[155,162,163,206,207]
UBC^®^ Rapid IDL Biotech	Cytokeratin (CK) 8/18 fragments Soluble marker POC immunoassay	36–79 50–59	88–92 82–86	[155,206]
Survivin Fujirebio Diagnostics Inc.	Anti-apoptotic protein ELISA, bio-dot assay	36–64	93–98	[163,206,208,209]
CYFRA 21-1 Bio International; Roche Diagnostics	Cytokeratin 19 fragments Soluble marker ELISA	70–90 Low for low-grade tumors (~55)	73–86 67–71 Low in benign conditions, intravesical instillations	[155,163,206,211]
ONCURIA™	ANG, ApoE, ANG, CA9, IL8, MMP9, MMP10, PAI1, SDC1, VEGF1 Soluble marker Multiplex immunoassay False positives in hematuria can be reduced by urine centrifugation	79–92.2%	~80	[164,177,178,179]
Assure MDx MDx Heath USA	Mutations in FGFR3, TERT, and HRAS OTX1, ONECUT2, and TWIST1 promoter methylation	93–97 Claimed as highest diagnostic odds ratio; limited testing	83–86	[165,193,194] Not extensively evaluated
ADXBLADDER™	MCM5 transcript expression RT-qPCR	45–73 May rule out aggressive tumors with 99% negative predictive value	80–88	[190,191,192]
CxBladder™ Cxb CxbD CxbR Pacific Edge, Dunedin New Zealand	IGBP5, HOXA13, MDK, CDK1, CXCR2 transcripts Multiplex PCR Three tests with different objectives	91.1–100 Possible 35% to 39% reduction in cystoscopy Hematuria: 85	75–93%; surveillance 73.3% (hematuria)	[161,180,181,182,183]
XPERT© Bladder Cancer Monitor, XBCM Cephaid Sunnyvale, USA	CRH, IGF2, UPK1B, ANXA10, ABL1 transcripts Urine RT-qPCR linear discriminant analysis for determining test’s inference	58–86 Low-grade 40–78 High-grade: 86–92 Possible ~75% reduction in cystoscopies but misses ~25% of recurrence	73–91	[184,185,186,187,188,189]
Uromonitor^®^ Uromonitor-V2^®^, U-monitor, Porto, Portugal	TERT promoter (c.1-124C > T and c.1-146C > T) and FGFR3 (p.R248C and p.S249C) hotspot mutations, Hotspot alterations in three different genes (TERT, FGFR3, and KRAS) Urine DNA qPCR test	Uromonitor^®^: 73 Uromonitor-V2^®^: 93	93 85	[199,200]
Bladder EpiCheck Nucleix, Rehovot, Israel	Methylation profile of 15 genomic loci DNA qPCR assay in urine	64–90	82–88	[201,202,203,204,205]

POC: point of care.

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
