# Peer review of "Molecular Oncology of Bladder Cancer from Inception to Modern Perspective"

_cancers, 2022, doi:10.3390/cancers14112578_

Round 1

Reviewer 1 Report

This is a well written review but could be improved further:

In the abstract the authors refer to "divergent yet interrelated" molecular pathways for low and high grade bladder tumours.  However, in the review the pathways are considered separate.  It would be really interesting to know how they are "interrelated" and where grade 2 and intermediate risk NMIBCs fit into the 2 pathway model.  

Please provide references for the 50% and 80% recurrence rates for low and high grade bladder cancer (first paragraph, page 4).

Please provide references for the sentence beginning "lineage tracing" (first paragraph, page 5).

It is my understanding that chr 9 losses are common to low and high grade pathways.  Please provide references supporting the statement that chr 9 LOH distinguished between the two (page 6).

STAG2 mutations also show gender bias (page 6).

MDM2 is often amplified (not mutated).

Overall, the consideration of genomic changes is quite basic and phenomena like APOBEC mutagenesis, TMB and altered pathways could added.

Section 7 should be entitled "Transcriptomic....." rather than "Genomic......" 

It would be good to have the "forerunner gene" concept clarified/expanded.

In the single cell sequencing section (page 8) the authors appear to think that reference 60 shows that a single cancer cell can simultaneously express multiple subtypes - please re-word!

Page 10 - please provide a reference supporting the notion that 2/3 of high grade cases are MIBC.

Author Response

Manuscript ID cancers-1704254

Title: Molecular Oncology of Bladder Cancer from Inception to Modern Perspective

Soum D Lokeshwar , Maite Lopez , Semih Sarcan , Karina Aguilar , Daley S Morera , Devin M Shaheen , Bal L Lokeshwar * , Vinata B Lokeshwar *

Response to Reviewer 1’s comments:

This is a well written review but could be improved further:

  1. In the abstract the authors refer to "divergent yet interrelated" molecular pathways for low and high grade bladder tumours.  However, in the review the pathways are considered separate. It would be really interesting to know how they are "interrelated" and where grade 2 and intermediate risk NMIBCs fit into the 2 pathway model.  

Response: We thank the reviewer for this insightful comment. Based on the reviewer’s suggestion, we have revised the manuscript as follows:

The origin of low-grade tumors is traced to the intermediate cell layer [79-81].  Contrarily, high-grade  NMIBC arises from a different cellular and molecular pathway with early hyperplasia combined with severe dysplasia. In the two-pathway model, Tis was proposed as arising from dysplasia alone. Both high-grade NMIBC and Tis have a high propensity to invade and metastasize [28,79-83]. The two divergent pathways were also proposed to be interrelated, as both high-grade NMIBC and Tis have the potential to progress to MIBC [5,7]. (Page 11, line 3)

Cytogenetic studies and mapping of the molecular alterations have supported the divergent pathway model but have also supported molecular instability associated with disease progression. Cytogenetic alteration analysis showed a “temporal order” in the appearance of chromosomal imbalances with an increase in tumor grade and stage [96]. Molecularly it was proposed that BC arises from two cytogenetic pathways. One of these proposed pathways is initiated by chromosome 9 alterations followed by alterations in chromosomes 11 (-11p) and 1 (1q+). This pathway correlated with stage Ta-T2 tumors. The other pathway was proposed to be initiated by a gain of chromosome 7 (+7) followed by alterations in chromosome 8 loci (8p-; +8q) [7,96]. The evolving state-of-the-art molecular and profiling techniques have mapped the genomic atlas of BC. (Page 12, line 8).

Nevertheless, based on molecular heterogeneity among BC and its association with BC progression, it would be important to study how the intermediate grade BC (grade 2 per 1973 WHO classification) and intermediate risk NMIBCs fit into the two-pathway model.  The implementation of the 2004 WHO/ISUP system has been reported to either underdiagnose high-grade BC or over-diagnose BC as high-grade [153-155]. In this regard, molecular subtyping could be useful in accurately determining bladder tumor grade. This would have immediate clinical significance, as the management of patients with low- and high-grade tumors is vastly different. (Page 20, line 12).  

  1. Please provide references for the 50% and 80% recurrence rates for low and high grade bladder cancer (first paragraph, page 4).

Response: Based on the reviewer’s suggestion, we have elaborated this section and added the relevant references:

As was reported nearly forty years ago, bladder tumors demonstrate variable recurrence and progression rates [41,42].  Tumor recurrence rates of about 50% have been reported for low-grade tumors, most of which are stage Ta and have a low progression rate [41,43,44]. Recurrence rates of 57%, 67% and 77%, and 5-year progression rates of 7%, 26% and 46% have been reported for low, intermediate, and high-risk patients with NMIBC, respectively [45]. In this study, 92% of patients had high-grade NMIBC and underwent BCG treatment. In high-risk high-grade BC, 74.3% recurrence rate has been reported [46].. (Page 7, line 17).

  1. Please provide references for the sentence beginning "lineage tracing" (first paragraph, page 5).

Response: We now have added the relevant reference: The two-track model explains the divergent pathways of the development of papillary and non-papillary invasive tumors. This has been further supported by lineage tracing experiments in mouse models [78-81]. (Page 11, line 23).

  1. It is my understanding that chr 9 losses are common to low and high grade pathways.  Please provide references supporting the statement that chr 9 LOH distinguished between the two (page 6).

Response: We thank the reviewer for this comment. We now have revised and expanded this write-up:

LOH on chromosome 9 (9p, 51%; 9q, 57%; even up to 70%) is both an early, and the most common genetic alteration identified in BC, and it is observed in all stages and grades [87-93]. The loss-of-function genetic alterations (mutations and loss of heterozygosity) in tumor suppressor genes mapping within chromosome 9p (CDKN2A/p16INK4a/p14ARF; CDNKN2B/MTS-2)/p15INK4b) and 9q (PTCH1, TSC1 and DBC1) loci were discovered to be common in NMIBC.  (Page 11, line 26).

“Cytogenetic studies and mapping of the molecular alterations have supported the divergent pathway model but have also supported molecular instability associated with disease progression. Cytogenetic alteration analysis showed a “temporal order” in the appearance of chromosomal imbalances with an increase in tumor grade and stage [96]. Molecularly it was proposed that BC arises from two cytogenetic pathways. One of these proposed pathways is initiated by chromosome 9 alterations followed by alterations in chromosomes 11 (-11p) and 1 (1q+). This pathway correlated with stage Ta-T2 tumors. The other pathway was proposed to be initiated by a gain of chromosome 7 (+7) followed by alterations in chromosome 8 loci (8p-; +8q) [7,96]. The evolving state-of-the-art molecular and profiling techniques have mapped the genomic atlas of BC. (Page 12, line 8).

  1. STAG2 mutations also show gender bias (page 6).

Response: Thank you so much. We have now modified the sentence as follows: Mutations in both genes show gender bias and are twice more common in women with low-grade NMIBC than in men [84,100,101].  (Page 13, line 7).

  1. MDM2 is often amplified (not mutated).

Response: Based on the reviewer’s suggestion, we have modified the sentence as follows: MDM2 gene amplification has been reported in BC [109,110]. Although MDM2 mutations are less common in BC, the Cancer Genome Atlas (TCGA) mutational analyses studies showed that mutations in TP53 and MDM2 genes are mutually exclusive [111-113]. (Page 14, line 16).

  1. Overall, the consideration of genomic changes is quite basic and phenomena like APOBEC mutagenesis, TMB and altered pathways could added.

We thank the reviewer for this comment. We have added a whole new section with numerous references on genomic alterations/tumor mutation burden in BC. Each of the topics suggested by the reviewer has been recently reviewed, including in this journal (Cancers). The present review is an attempt to give a reader an overview of the field of the molecular oncology of bladder cancer, and the clinical signficance of this research. We hope the reviewer will agree with our view.

Genomic alterations/Tumor mutation burden in BC

It is well known that every tumor harbors DNA alterations/mutations. One of the first comprehensive attempts to catalog a consensus of cancer genes showed more than 1% of the genes in the human genome contribute to cancer [160]. Cataloging of cancer mutations was initiated when 4,938,362 mutations from 7,042 cancers were documented and the study described more than 20 distinct mutational signatures. The study described that in addition to the genome-wide mutational signatures, hypermutations localized in small genomic regions, “Kataegis” are found in many cancers [161].  COSMIC, the Catalogue Of Somatic Mutations In Cancer (https://cancer.sanger.ac.uk), is a widely used, and heavily cited comprehensive resource for exploring the effect of somatic mutations in human cancer [162]. The most recent COSMIC v86 version includes nearly 6 million mutations in coding regions across 1.4 million tumor specimens, which are curated from over 26,000 publications. COSMIC includes all somatic mutations that promote cancer growth and progression and include non-coding mutations, gene fusions, copy number variants and drug-resistance mutations [162]. However, COSMIC is a “live document” as novel mutations are found across all chromosomes in various cancer types and tumor specimens in a random fashion. The random nature of the mutations across the entire human genome (i.e., no physical association) indicates that although there are “driver mutations”, the majority of the “coordinated regulation” may be at the transcriptional and epigenetic levels [163] 

Studies on the TCGA dataset have revealed that BC has a higher mutation rate exceeded probably only by lung cancer and melanoma [164,165].  About 65% of mutations found in cancers are mediated by APOBEC (apolipoprotein B mRNA-editing enzyme, catalytic polypeptide-like) family of cytidine deaminases. APOECs affect many signaling pathways involved in cancer growth and progression. Mutations in chromatin-modifying genes and transcription factors are also common in BC [164]. The human genome contains several homologous APOBECs and APOBEC1 and APOBEC2 account for most of the point mutations [164]. Cytidine deaminases convert cytosine to uracil and like the most studied “activation-induced cytidine deaminase”, the APOBEC class of enzymes is involved in both innate immunity and RNA editing [166]. Upregulation of APOBEC(s) in tumors enhances Cï‚®T transitions and Cï‚®G transversions with preference to 5’TCW (W=A/T) or TCG DNA trinucleotides. Genomic deamination events lead to cytosine mutagenesis clusters (kataegis) causing chromosomal aberrations such as translocations [165-168]. These simultaneous mutation clusters found within the same strand are generated by base alkylation of long single-strand DNA, which forms at double-strand DNA breaks/chromosomal arrangement break points and replication forks. Therefore, APBOEC family induces hypermutations by causing multiple simultaneous changes in randomly created ssDNA [169]. In BC, a mutational pattern is altered based on APOBEC enrichment. APOBEC-high tumors have a significantly higher frequency of mutations in the TP53/RB1 pathway, DNA damage response and chromatin-modifying genes, whereas, APOBEC-low tumors show a higher frequency of KRAS and FGFR3 mutations [170]. In the TCGA-2014 BC dataset with 131 urothelial carcinoma specimens (all high-grade MIBC), 51% showed Tp*C->(T/G) mutations and high expression of APOBEC3B [113]. An independent report confirmed the upregulation of APOBEC3 activity in BC [171]. In the context of 238 TCGA BC specimens, two variations of APOBEC mutation signatures were found. One consists of Cï‚®T transitions (TCW motif) accounting for 17% of single nucleotide variations, and another accounting for 48% of single nucleotide variations (both Cï‚®T and Cï‚®G mutations) [164]. In the TCGA-2017 dataset results on 412 specimens were reported [130]. In that study APOBEC mutation signature was observed in 67% of the tumors and mutagenesis correlated with the expression of APOBEC-3A and APOBEC-3B [130]. Most of these mutations had a clonal origin, suggesting that APOBEC-mediated mutations occur early in the development of BC. It is noteworthy that Patients with high APOBEC-signature mutagenesis and high mutation burden showed significantly higher (75%) 5-year survival probability. Better survival was also seen in subsets defined by high mutation burden or high APOBEC-signature mutation load [130]. A study by Glaser et al included 395 specimens from the TCGA dataset and showed that APOBEC-high tumors have better overall survival than those with APOBEC-low tumors [170].

As some of us have reported earlier, the TCGA-2017 lists 436 specimens. Among these 23 are normal bladder, one “metastasis”, five with missing RNA-Seq data, 5 NMIBC (T0 N0 M0: 1; T1N0 M0: 3; TX: 1) and 402 listed as MIBC. Among the 402 MIBC specimens, 21 are “low-grade” tumors [142]. Therefore, the 412 specimens reported in the TCGA-2017 study or 395 specimens reported in the Glaser et al study contain both low-grade tumors (which are rarely MIBC), and/or NMIBC specimens [59,130,170]. Since APOBEC mutations likely occur early in the development of BC, if would be clinically significant to reanalyze the data on the 412 or 395 specimens to determine if low-grade and high-grade tumors have an equal frequency of APOBEC-mediated mutations and are the mutations similar in nature. Furthermore, it remains to be determined if the correlation of APOBEC mutations with better survival also depends on the presence of the low-grade tumors in this dataset. Notably, in the Beijing Genomics Institute (BGI) cohort and in a subset of MIBC specimens, no significant correlation was observed between survival and APOBEC levels [59,170]. In the broader context, such an analysis could reveal further proof of the divergent molecular pathways for the development of low- and high-grade tumors, and could further elucidate whether these pathways are interrelated (i.e., similar mutation pattern or dissimilar mutation frequency/pattern) and how they drive the clinical course of BC.

Mutational analysis of  5,243 cancer genome sequences (4,633 exomes and 610 whole genomes) of smoking-related cancers has revealed a signature (signature 5) that harbors inactivating mutations in ERCC2 which encodes a component of nucleotide excision repair.  The signature was found to be associated with BC and was higher among smokers compared to non-smokers [172]. This signature is characterized by T→C and C→T mutations with a transcriptional strand bias for T>C mutations [172]. Analyses of the TCGA dataset for mutations among smokers and non-smokers with MIBC found only one gene that was mutated – GPR15 and it was found to be up-regulated in both luminal- and basal-subtype tumors from smokers [173]. It is noteworthy that in the TCGA dataset, all low-grade tumors (listed as MIBC), are of the luminal subtype. Therefore, finding a smoking-related mutational signature using this dataset may not reveal how smoking changes the mutational landscape during BC development [142]. Tumors with somatic mutations in  ERCC2 and other nucleotide excision repair genes (signature 5*) were also found to be associated with a higher mutation burden in urothelial tumors compared to those with wild-type ERCC2 and other nucleotide excision repair genes [59,142]. A composite mutational signature of ERCC2 deficiency was found to correlate with cisplatin response in patients with advanced BC [174]. ERCC2 missense mutations are also enriched in primary MIBC (treatment naïve) compared to secondary MIBC (resulting from NMIBC progression) and is possibly a predictor of increased sensitivity to cisplatin-based therapy in the neoadjuvant setting [175,176].

As reviewed recently by Cooley et al mutational using the TCGA-2017, MSKI and BGI datasets have reported enrichment of mutations in MIBC in various cellular pathways. Among these, mutation frequency (≥ 50%) is found in cell cycle-related genes (e.g., TP53, RB1, CDNK1A, CDNK2A), chromatin regulatory genes e.g., (KTM2D, KDM6A, STAG2, ARID1A, KTM2C, EP300, CREBBP, BRWD1), and in RAS/PI3-K pathway genes [59]. A slightly lower (20% - 30%) but significant mutation frequency has been noted in the chromatin remodeling genes (reviewed in [59]). Whole organ mapping for studying the locoregional changes has revealed DNA methylation alterations and their association with the initiation of field-effect in BC. The field-effect likely occurs concurrently with sub-clonal low-allele frequency mutations and a small number of DNA copy alterations. The study found an initiating mutation in ACIN1 in the normal mucosa with 21 additional mutations associated with cancer progression [39]. The role of ACIN1 in bladder carcinogenesis remains to be evaluated. The mutational landscape studies have advanced the molecular understanding of the field-effect in bladder carcinogenesis and the pathways involved in BC progression. Multi-omic approaches involving genomics, transcriptomics, epigenomics and proteomics analyses confirm genomic instability in MIBC and certain gene signatures associated with poor response [177].

Along with the genomic/transcriptomic approaches, metabolomic analyses of urine specimens and the availability of the Human Metabolome Database, and the bulk epigenome analyses approaches have identified altered pathways in BC, some of which are likely associated with bladder carcinogenesis, growth, and progression to MIBC. Recent reviews describe these discoveries [178-182]. Currently the majority of the “omic” studies at all levels (gene, epigenetic, transcript, protein, metabolites) are correlative. These approaches should pave the way for the much needed  systematic mechanistic and functional studies for developing personalized targeted and synthetic lethality treatment regimens and clinically significant prognostic markers for BC.  (Page 21, line 8).

  1. The section should be entitled "Transcriptomic“ rather than "Genomic“

As per the reviewer’s suggestion, we have changed the title of the section: “Transcriptome profiling and the quest for personalized medicine“. (Page 15, Line 7).

  1. It would be good to have the "forerunner gene" concept clarified/expanded.

We agree with the reviewer. The concept of “Forerunner gene“ together with whole organ genome mapping is an innovative, clinically significant concept and is mechanistically testable. It can explain genetic alterations that drive the development of low and high-grade bladder cancer and progression. The whole genome bulk sequencing studies and molecular subtyping overshadowed this innovative concept. We hope that our review article will stimulate research collaborations to discover driver genes and then to conduct mechanistic and functional validation studies on forerunner genes – the genes that drive bladder cancer. We have modified the section as follows:

Strategies such as whole organ genome mapping were/are employed to identify driver mutations in “forerunner genes” and provide evidence for the clonal origin of tumors [19,39,102]. Czerniak’s group proposed the concept of forerunner genes fifteen years ago. This concept was based on whole-organ histologic and genetic mapping (WOHGM) studies for identifying clonal hits associated with growth advantage, thus tracking the development of human bladder cancer from occult in situ lesions [20,39,103]. The initial study identified three “major waves” of genetic alterations that reflect the evolution of normal urothelial cells into cancer cells by possibly acquiring a growth advantage resulting from these genetic alterations. These changes were shown to map to six regions at 3q22–q24, 5q22–q31, 9q21–q22, 10q26, 13q14, and 17p13, and were proposed to be “critical hits” that drive BC development [19]. Among these,chromosome 13q34 locus encompassing RB1 was associated with Tis [20]. MTAP gene on chromosome 9p21 could be another forerunner gene, as its homozygous loss is associated with poor survival [104]. However, since LOH at chromosome 9 is an early event in the development of BC, and the locus also harbors p16 (CDKN2), it emphasizes the need for functional studies to define if a genetic alteration is genuinely a forerunner gene that drives BC development or if it is a bystander [104,105]. The concept of the forerunner gene was and still is an attractive concept. It could explain an initiator driver genetic alteration(s) that leads to the development of low- and high-grade BC, malignant progression of MIBC, treatment resistance and synthetic lethality. This concept possibly gave way to molecular subtypes using bulk “-omic” approaches. However, the bulk -omic approaches are designed to reveal associations between any alteration (genes, transcripts, proteins, or metabolites) and BC (especially MIBC) and not necessarily the drivers of cancer development and progression, as the forerunner gene concept proposed. Therefore, functional and mechanistic investigations beyond the correlative studies are needed to identify the true “drivers” or forerunner  - genes, transcript variants, proteins and/or metabolites -  as novel  targets for developing precision treatments or prognostic markers for patients with advanced BC. (Page 13, line 11).

  1. In the single cell sequencing section (page 8) the authors appear to think that reference 60 shows that a single cancer cell can simultaneously express multiple subtypes - please re-word!

Respone: We thank the reviewer for correcting our inadvertant error. The sentence has been rewored as follows: Single-cell sequencing has identified that a single cancer cell can simultaneously express multiple subtypes [67]. (Page 18, line 14).

  1. Page 10 - please provide a reference supporting the notion that 2/3 of high grade cases are MIBC.

We have clarified the statement: High-grade bladder tumors have high invasive potential. Approximately 25% of patients with BC present with a tumor invading the muscle layer of the bladder wall (T2), perivesical fat (stage T3) or the adjacent organs (stage T4; Figure 1) [38].  (Page 6, line 7).

Thank you again for your insightful comments which have improved the scientific merits of this review.

Reviewer 2 Report

The authors review the molecular mechanisms regarding the bladder cancer, and the description is in-depth and the figures are well illustrated. I do not find any errors in the text. 

 The review well summarised a number of genes that are related to the outset of the bladder cancer initiation and progression, which had been poorly understood until recently. and I thought that this kind of summary can hardly be found elsewhere. They cited a good number of original and recent articles and I thought this review is useful for a number of researchers who are working on cancer, including myself.  However, after I found another reviewer’s comments, I have been convinced with this reviewer’s comments. I therefore recommend minor revision as the decision according to the first reviewer’s careful comments.

Author Response

Manuscript ID cancers-1704254

Title: Molecular Oncology of Bladder Cancer from Inception to Modern Perspective

Soum D Lokeshwar , Maite Lopez , Semih Sarcan , Karina Aguilar , Daley S Morera , Devin M Shaheen , Bal L Lokeshwar * , Vinata B Lokeshwar *

Reviewer # 2:

….. I thought that this kind of summary can hardly be found elsewhere. They cited a good number of original and recent articles and I thought this review is useful for a number of researchers who are working on cancer, including myself. However, after I found another reviewer’s comments, I have been convinced with this reviewer’s comments. I therefore recommend minor revision as the decision according to the first reviewer’s careful comments.

Response: We thank the reviewer for recognizing the timeliness of this review article. We cannot agree more that this type of summary can hardly be found elsewhere. In fact, because we could not find such a summary, we undertook this project. Reviewer # 1’s comments promoted us to include new sections, especially the one on genomic profiling of bladder tumors. As a result, the revised manuscript has 8,988 words, compared to 7,035 words in the original manuscript. We hope the revised review will provide readers a critical review and understanding of the seminal advances that have been made in the basic, translational, and clinical arenas of the molecular oncology of bladder cancer within the last forty years.
